# Fungal Hyphae on the Assimilation Branches Are Beneficial for *Haloxylon ammodendron* to Absorb Atmospheric Water Vapor: Adapting to an Extreme Drought Environment

**DOI:** 10.3390/plants13091233

**Published:** 2024-04-29

**Authors:** Xiaohua Wang, Honglang Xiao, Lei Pang, Fang Wang

**Affiliations:** 1Key Laboratory of Ecohydrology of Inland River Basin, Water and Soil Resources Research Office in Cold and Arid Regions, Northwest Institute of Eco-Environment and Resources, Chinese Academy of Sciences, Lanzhou 730000, China; 2Agronomy College, Gansu Agricultural University, Lanzhou 730070, China

**Keywords:** fungal hyphae, assimilation branches, *Haloxylon ammodendron*, atmospheric water vapor absorption, drought adaptability

## Abstract

Research on endophytic fungi in desert plants, particularly the epiphytic or endophytic fungi of leaves, remains limited. In the extremely arid regions of northwest China, the ultra-xerophytic desert plant *Haloxylon ammodendron* harbors white fungi on its assimilating branches during autumn. The hyphae of these fungi intertwine, both internally and externally, comprising superficial, bridging, and endophytic types. The superficial hyphae attach to the surface of the assimilating branches and continuously grow and intersect, forming a thick layer of felt-like hyphae. This thick, felt-like layer of hyphae facilitates the adsorption of atmospheric water vapor on the surface of the hyphae or the assimilating branches, allowing *H. ammodendron* to capture atmospheric moisture, even under low humidity. Some superficial hyphae penetrate the cuticle into the epidermis, becoming bridging hyphae, which can rapidly transport water from the outside of the epidermis to the inside. The endophytic hyphae shuttle within the epidermis, achieving rapid water transfer within the epidermis of the assimilating branches. The presence of these three types of hyphae not only enables the assimilating branches of *H. ammodendron* to achieve rapid water absorption and transmission, but also facilitates the uptake of atmospheric water vapor under low humidity conditions. We discuss the mechanism by which the hyphae promote water absorption from the perspectives of hyphal composition, the formation of felt-like structures, and environmental conditions. We consider the presence of fungal hyphae on the surface of the *H. ammodendron* assimilating branches as an inevitable ecological process in arid environments. This study provides important theoretical insights into the mechanisms underlying the strong drought resistance of desert plants in extremely arid regions and offers strategies for desertification control.

## 1. Introduction

Drought and salinization are the most typical desert environmental factors, as well as the most important abiotic stress factors restricting plant growth. Plants have evolved their own unique response mechanism to stress in a desert environment over a long period of time. Symbiosis with microorganisms is an important strategy used by desert plants to cope with extreme environments. Driven by common interests, plants and microorganisms have evolved into a community of joint stress-resistance. Plants provide environmental buffer zones for microorganisms to maintain micro-ecological stability, and microorganisms jointly cope with stress by stimulating plants to produce various enzymes and osmotic regulatory substances [1,2]; This collaboration also helps desert plants respond to stress by plant hormones regulation, nitrogen fixation, and phosphorus dissolution, improving the nutritional status of the host plants, as well as enhancing photosynthesis and antioxidant enzyme activities [3,4,5]. Endophytes of desert plants can improve the drought resistance of their hosts. Endophytes isolated from desert plants can secrete auxin, iron-producing vector, ACC deaminase, and other substances, improving the water absorption capacity of plants by promoting the development of plant roots, thus improving the survival of the host [3].

The phyllosphere of plants, which includes both the leaf surface and its interior, is an important part of the microbial community, including epiphytic fungi inhabiting the leaf surface and endophytic fungi living in the leaf without symptoms. With high species diversity, it plays an important role in ecosystem function [6,7,8,9,10,11]. However, the current research on endophytic fungi in plants is mainly focused on root endophytic fungi, and there are few studies on foliar epiphytes and endophytic fungi. Compared to many studies on plant rhizosphere fungi, studying the interaction between desert plants and the foliar epiphytic or endophytic fungi of leaves is particularly urgent and important for the stability and restoration of desert ecosystems, as the leaf or assimilating branch tissues are the central players in photosynthesis and energy metabolism.

*Haloxylon ammodendron* (C.A. Mey.) Bunge (Chenopodiaceae) is a dominant xerophytic woody species in the arid desert of Asia [12] that is resistant to drought, high temperature, salt and alkali, and wind erosion. It plays an irreplaceable ecological role in sand prevention and fixation, desertification mitigation, and ecological security maintenance, and it is known as the guardian of desert [12,13,14,15,16]. Studies have found that *H. ammodendron* exhibits many morphological characteristics to adapt to drought [17,18], such as a deep root and leaves which degenerate into scales. Transpiration and carbon assimilation occur on assimilating branches, and assimilating branches are sensitive to changes in water input in terms of physiological properties [12,19], and the assimilation branches, with special structures such as water-absorbing tissues and crystals, are considered to contribute to drought tolerance [20], but this has only been speculated and has not been confirmed.

In the field investigation, we observed that in the semi-arid, arid, or extreme arid areas of China, the epidermis of the assimilation branches of *H. ammodendron* was smooth in early spring or early and midsummer, and there is no presence of white hypha (Figure 1A,B). In the extreme arid areas, with extremely poor water conditions, the epidermis of the assimilation branches of *H. ammodendron* began to grow hyphae in late summer and late August (Figure 1C–E). Its surface is covered with a thick felt-like layer of white fuzzy hyphae (Figure 1F,G).

The literature suggested that in the thick white fuzzy hyphae fungus growing on the assimilated branches of *H. ammodendron* is *Leveillula saxaouli* (Sorok.) Golov., belonging to *Leveillula* genus, Erysiphaceae family [21], the conidium are thin and erect, growing from the stomata; the primary conidium are cylindrical, expanding into rings near both ends, and the tip becomes sharp; the secondary conidium conidia are also cylindrical, with both ends obtuse; the cleistothecium are spherical or oblate, initially pale yellow, then brown, and began to mature in early September [21,22]. This white powder fungus *L. saxaouli* overwinters on infected branches with cleistothecium. In spring, the cleistothecium sprouts and releases ascospores to infect new branches, which is the primary infection source. Conidium is formed in the white powder layer of the infected branches in the same year, which is spread by wind and repeatedly infects the assimilation branches. In August, it enters the expansion and spread period [22,23].

The phenomenon of this thick felt-like layer of fuzzy hyphae growing on the assimilation branches of *Haloxylon* is very widespread in arid areas of China, especially in extreme arid areas. In the whole of Xinjiang, 90–100% of *Haloxylon* is covered with felt-like white hyphae, 100% is covered in Qiemo County, and 90% is covered in Dengkou County and in Ejina Banner of Inner Mongolia [21]. These places are particularly dry, and rainfall is very low. The average annual rainfall is only 20 mm in Qiemo County and less than 40 mm—measuring only 7 mm in some years—in Ejina Banner. Under such arid conditions, the assimilation branches of *H. ammodendron* are covered with felt-like white hyphae, and this phenomenon is also widespread. Is this felt-like white hyphae structure a specialized structure formed by the interaction between *H. ammodendron* and fungi in order to better adapt to the arid environments?

In recent years, the study of vegetation water resources in arid areas has become a hot spot in regards to ecohydrological research. An increasing number of studies assert that atmospheric water vapor, fog, dew, and condensate directly absorbed by the above-ground parts of plants have become important water sources for arid plants to grow or escape drought [24,25,26,27,28,29]. It has been reported that many xerophytes or halophytes absorb and utilize atmospheric water vapor through their above-ground leaves or stems, enabling them to survive in arid habitats [30,31,32,33,34]. According to recent studies, the structure and composition of the plant epidermis is a decisive factor affecting the interaction between the plant surface and atmospheric water vapor [35,36,37], micro-nano level geometry (or roughness), and chemical composition of the epidermal surface determine the wettability, adhesion or repulsion, and permeability of water droplets [38,39].

The felt-like hyphae structure on the assimilation branches of *H. ammodendron* directly changes the structure and composition of the interaction between the surface of the assimilation branch and atmospheric water vapor. In addition, our previous studies have shown that *H. ammodendron* can absorb unsaturated atmospheric water vapor using assimilating branches. Therefore, we speculated that these felt-like hyphae structures growing on the surface of the assimilation branches might help *H. ammodendron* absorb atmospheric water vapor and enable it to survive in extremely arid areas.

To verify this hypothesis, we conducted atmospheric fluorescence humidification experiments on *H. ammodendron* plants with hyphae-covered branches and those without hyphae in Ejina Banner, evaluating the interaction between hyphae and the surface of the assimilating branches, as well as the impact on atmospheric water vapor absorption. This research is of significant importance for understanding the extremely strong drought resistance mechanism of *H. ammodendron*, and also for providing theoretical guidance for the cultivation management of *H. ammodendron*.

## 2. Results

### 2.1. Fungal Infiltration Law, Morphology, and Mycelium Structure Characteristics

The thick, white, felt-like hyphae of the fungus growing on the assimilation branches of *H. ammodendron* appeared as yellowish (Figure 2A,B) edematous patches (Figure 2C) at the initial stage of immersion, and a flocculent white powder layer appeared on the site of discoloration a few days later. Over time, the white powder layer gradually thickened and became felt-like, and at most, the white powder covered the entire assimilation branches (Figure 1F); in the later stage, yellow-brown to brown spots appeared in the white hyphae layer (Figure 2A,D), as the cleistothecium of the fungus. The cleistothecium was spherical or oblate, with a diameter of 15~26 μm (Figure 2E,F), buried in the hyphae, and began to mature in early September, with filamentous appendages (Figure 2F). The conidium was thin, erect, and grew out of the stomata (Figure 2G), with a size of (17~33) × (5~8) μm. The primary conidium was cylindrical, with a pointed tip (Figure 2H), and the secondary conidium was cylindrical, with obtuse ends (Figure 2H).

The assimilation branches of *H. ammodendron* in extremely arid areas are covered with white fungal hyphae in autumn (Figure 3). Under natural conditions, the hyphae are not particularly clear under either bright or fluorescent fields (Figure 3A,B) and emit a light gray fluorescence under fluorescence microscopy (Figure 3B). After FB fluorescence humidification, the hyphae absorbed a large amount of fluorescent reagents and fluoresced blue (Figure 3C–E). The primary conidium is cylindrical, with a pointed tip (Figure 3A,B). The secondary conidium has a blunt tip (Figure 3A). The conidium stands erect and grows out of the stomata (Figure 3G). The cleistothecium is buried in the hyphae, and is oblate and spherical (Figure 3C); there are multiple asci in the cleistothecium. One part of the hyphae is endogenous, and the other part is superficial. The hypha is a long filament composed of tubular structures with a transverse septum. It snakes along the surface of the cuticle or is embedded in the epidermis, traversing the cuticle into the epidermis multiple times (Figure 3D). A large number of hyphae repeatedly interweave on or near the epidermis and spin into a thick, felt-like hyphae structure (Figure 3I).

The hyphae pass through the cuticle and enter the space between the epidermal cells and the hypodermis cells, but do not enter the cytoplasm. Its cross section is irregular, the diameter of the tube is 2.7–6.26 µm, and the wall thickness is 0.7–1.1 µm (Figure 3G,H arrow points). A large number of hollow and irregular circular cross sections of the hyphae can also be seen near the outer surface of the epidermis (indicated by the arrow in Figure 3I). A large number of hyphae interweave near the epidermis to form thick, felt-like structures with a thickness of about 180 µm (Figure 3I).

### 2.2. Comparison of Water Content and Water Potential of Assimilation Branches, with Hyphae and without Hyphae, under Ultrapure Water Humidification

Figure 4 shows the comparative changes in the water content and water potential of assimilated branches, with and without hyphae, during the humidification with ultrapure water. Under the temperature and humidity conditions in Figure 4A,B, the relative humidity was increased to 85–95%. With the extension of humidification, the water content and water potential of the hyphae-containing and hyphae-free assimilation branches continuously increased (Figure 4C,D). The water content of the hyphae-containing assimilation branches increased from 73.46% ± 0.08% to 76.07% ± 0.08%, while the water potential could not be measured, i.e., it was less than −7 MPa (the minimum range of the water potential instrument is −7 MPa, so we used −7 MPa for our calculations) to −4.34 MPa; the water content of the hyphae-free assimilation branches increased from 74.19% ± 0.09% to 75.94% ± 0.08%, and the water potential increased from −5.78 ± 0.09 MPa to −4.48 ± 0.15 MPa. Both increased significantly in the first stage (from the beginning of humidification to 1 h, i.e., 0–1 h) and the second stage (from 1 h to 2 h, i.e., 1–2 h) (Figure 4C,D). The water content and water potential of assimilation branches with hyphae increased by 1.97% and 2.15 MPa, respectively; the water content and water potential of assimilation branches without mycelium increased by 1.13% and 0.84 MPa, respectively.

From the third stage (from 2 h to 3 h, i.e., 2–3 h) to the fourth stage (from 3 h to 13 h, i.e., 3–13 h), the range of increase of both decreased. In the fourth stage, the increase rate of water content and water potential of both was the lowest, indicating that after 3 h of water absorption under this condition, both tended to be saturated. However, in the fourth stage, the increase rate of water content of the hyphae-free assimilated branches was higher than that of the hyphae-containing assimilated branches, indicating that the assimilation branches with hyphae tended to become saturated earlier than those without hyphae.

In the whole humidification process, the increase in the assimilation branches with hyphae was larger than that in those without hyphae from the first stage to the third stage, which indicated that under the temperature and humidity conditions of Figure 4A,B, the assimilated branches with hyphae absorbed atmospheric water vapor faster than those without hyphae in the first 3 h of humidification at relative humidity of 85–95%; especially in the first stage, the difference reached a significant level (water potential, *p* < 0.01; water content, *p* < 0.05) (Figure 4E,F). This suggests that the hyphae can promote the absorption of atmospheric water vapor using the assimilating branches of *H. ammodendron*.

### 2.3. Water Absorption Sites and Characteristics of Assimilating Branches with Hyphae

Figure 5 illustrates a comparison of fluorescence humidification between *H. ammodendron* assimilation branches, with and without hyphae, in the same frame, conducted in the extremely arid area of Ejina Banner. The hyphae on the surface of *H. ammodendron* on the assimilation branches were identified as the water absorption site by fluorescence microscopy, and the hyphae could promote the absorption of atmospheric water vapor through the assimilation branches of *H. ammodendron*.

After about 0.5 h of fluorescence humidification at 60–65% (Figure 5A), it can be seen that hyphae on the surface of the assimilation branches emit blue fluorescence (Figure 5B(a)). With the extension of humidification time, the intensity of blue fluorescence emitted by the hyphae becomes increasingly stronger, and progressively more water is absorbed by the hyphae (Figure 5B(b)). At about 3 h of fluorescence humidification at 80–90%, the entire epidermal hyphae emitted strong blue fluorescence (Figure 5B(c)), and the hyphae absorbed sufficient water. These results indicate that the hyphae serve as the water absorption site of the assimilation branches of *H. ammodendron*, and that the hyphae have a strong ability to absorb water.

Figure 5B(d–f) depicts a comparison of water absorption between areas with hyphae and areas without hyphae on a complete cross-section of an assimilation branch of *H. ammodendron* after about 4.5 h of fluorescence humidification. Under the same humidification conditions, both the hyphae and the cross-section at the sites with hyphae emitted strong blue fluorescence (Figure 5B(e)), while that of the areas without hyphae emitted weak blue fluorescence (Figure 5B(f)). This indicated that the strong blue fluorescence emitting in Figure 5B(e) was caused by the absorption of atmospheric water vapor of the hyphae. Due to the lack of absorbing water, the assimilation branches of this site, without hypha emitted blue fluorescence, was not strong. This well demonstrated that the hyphae on the surface of the assimilating branches of *H. ammodendron* could absorb atmospheric water vapor and promote the absorption of atmospheric water vapor by the assimilating branches.

Figure 6 showed the fluorescence comparison of the cross sections, with and without hyphae, under fluorescence humidifying, as shown in Figure 5A, in the Ejina Banner Region. When humidified at 60–65% fluorescent relative humidity for about 0.5 h, the assimilating branches with hyphae (including the hyphae and the epidermal and mesophyll cells below the hyphae) emitted strong blue fluorescence (Figure 6B), while the assimilation branches without hypha only emitted a small amount of blue fluorescence (Figure 6A). The assimilation branches with hyphae emitted significantly stronger blue fluorescence than those without hyphae, indicating that hyphae could absorb water, and that the absorbed water had partially transmitted to the mesophyll cells, resulting in a stronger blue fluorescence emitted by the mesophyll cells. However, the assimilation branches without hyphae have no hyphae on the surface to promote water absorption, so the amount of water entering the epidermis and the mesophyll cells was lower, and the blue fluorescence emitted was weak.

When the relative humidity of fluorescence humidification was increased to 85–90% and the humidification time was extended to about 1.5 h (Figure 5A), the epidermal and hypodermal cells of the assimilation branches with hyphae emitted strong blue fluorescence, and some mesophyll cells and water-storage tissue cells also emitted blue fluorescence, indicating that a large amount of water absorbed by the hyphae had entered the epidermis and hypodermis, and some had entered the mesophyll cells and water storage tissue, while the assimilating branches without hyphae only emitted blue fluorescence from the epidermis and hypodermal cells, and some mesophyll cells emitted a small amount of blue fluorescence, indicating that the water absorbed by the assimilating branches without hyphae only entered the epidermis and hypodermis, and very little water entered the mesophyll cells (Figure 6C,D).

As shown in Figure 5A, under the humidification method, the high humidity of RH 85–90% extended to 4.5 h, all the mesophyll cells of the assimilation branches with hyphae emitted strong blue fluorescence, and the water storage tissue also emitted blue fluorescence, while the assimilation branches without hyphae only emitted strong blue fluorescence in the epidermis and hypodermal cells, and most of the mesophyll cells still emitted red fluorescence. This indicated that a large amount of water entered the mesophyll cells and water storage tissue of the assimilation branches with hyphae, while water absorbed by the assimilation branches without hyphae had mainly reached only the lower epidermis, and part of the water reached mesophyll cells, but no water reached the water-storing tissues (Figure 6E,F).

From the above analysis, it was observed that under the fluorescence humidification at 0.5 h, 1.5 h, 3 h and 4.5 h, the assimilation branches with hyphae emitted stronger blue fluorescence than those without hyphae under the same conditions, and the water absorbed entered into deeper parts of the assimilation branches. These results showed that the assimilation branches with hyphae exhibit a stronger water absorption ability and a faster transmission speed, and also indicated that hyphae can promote the absorption of atmospheric water vapor through the assimilation branches of *H. ammodendron*.

### 2.4. Transcuticular Transport of Hyphae Promotes Water Absorption through the Assimilating Branches of Haloxylon ammodendron

In September 2019, we conducted a fluorescence humidification experiment on two types of *H. ammodendron* plants in the Ejina Banner area, one with hyphae on the surface of assimilation branches, and the other without hyphae, which were placed in the same glass frame humidifier to compare their differences in regards to water absorption pathways (Figure 7). It can be seen from Figure 7 that the cuticle did not emit blue fluorescence, but epidermal cells emitted blue fluorescence for the assimilation branches with hyphae (Figure 7A,C,E), while in the assimilation branches without hypha, in addition to the epidermal cells emitting blue fluorescence, the cuticle also emitted strong blue fluorescence (Figure 7B,D,F). This indicated that the hyphae on the epidermis of the assimilating branches directly transferred water from the outside of the epidermis to the inside of the epidermis, without passing through the cuticle, while the water on the surface of the assimilating branches without hyphae had to pass through the cuticle to enter the dermis.

Secondly, Figure 7 also shows that the transfer rate of water by the hyphae, directly from outside the epidermis to inside the epidermis, without passing through the cuticle, was faster than that of the water passing through the cuticle into the epidermal cells from the stomata or vertical cell wall of epidermal cells. Figure 7A,C,E shows fluorescence images for 1.5 h of fluorescence humidification, while Figure 7B,D,F shows fluorescence images for about 3–4.5 h of humidification under the same humidification conditions, both of which indicated the transfer of water from outside to inside the epidermis. In other words, it took only 1.5 h for the hyphae to transfer water across cuticle from outside the epidermis to the hypodermal cells and even to the mesophyll cells, while it required 3 to 4.5 h to transfer water passing through the cuticle into the hypodermal cells from stomata or the vertical cell wall of the epidermal cells. In addition, it can be seen from Figure 7A,C,E, with hyphae on the surface of the epidermis, that the hyphae passed through the epidermal cells in Figure 7A,E with stronger blue fluorescence (yellow arrow), but not in Figure 7C, which also strongly proves that hyphae pass through the cuticle and enter into the epidermal cells, which is conducive to water transmission in the assimilation branches.

From the above analysis, it could be seen that the fungal hyphae of the assimilation branches of *H. ammodendron* transfer water into the epidermis directly through hyphae, but not the water, extending through the cuticle. The leapfrog transport speed of the hyphae was much faster than the speed of water going through the cuticle. After the inhaled atmospheric water vapor entered the epidermis, the water transport mainly depended on the apoplast pathway. The water that entered the epidermal cells was first transferred laterally to the adjacent epidermal cells, and then longitudinally to the mesophyll cells, where it was stored in large quantities in the water storage tissues of the assimilation branches.

## 3. Discussion

### 3.1. Analysis of the Mechanism of Hyphae Promoting Water Absorption

The fusion structure of the endophytic hyphae and the epiphytic hyphae on the assimilating branches of *H. ammodendron* is beneficial for promoting the water absorption of the assimilating branches. The main reasons are as follows.

First, the components of the hyphae are hydrophilic. The cell wall of the fungal hyphae is mainly composed of chitin polysaccharides, which are polar molecules equivalent to water. Therefore, fungal hyphae are hydrophilic substances, making them prone to adsorb water molecules onto their surfaces. Compared to the hydrophobic cuticle layer of the assimilation branches, the chitin on the surface of the hyphae exhibits stronger hydrophilicity, allowing it to absorb water molecules more rapidly. This adsorption results in higher water content on the surface of the hyphae. Our experiment confirms this very well. When exposed to 60% atmospheric relative humidity for 0.5 h of fluorescence humidification, almost all the hyphae on the assimilation branches emitted blue fluorescence, indicating that a significant number of water molecules were adsorbed onto the hyphae. In contrast, there was only sporadic blue fluorescence on the surface of the assimilation branches without hyphae. In addition, the comparison of water potential before and after humidification from the assimilation branches of *H. ammodendron* growing hyphae can also clearly illustrate this problem. Before humidification, the water potential of the assimilation branches with hyphae could not be measured because it was too low, while the assimilation branches without hyphae could be measured. After humidification, the water potential with hyphae could be measured, and the measured water potential was higher than that without hyphae. On one hand, this indicates that the hyphae can quickly absorb atmospheric water vapor after humidification, thus increasing the water potential of the assimilation branches. On the other hand, it also suggests that the hyphae significantly reduce the water potential on the surface of the hyphae, which is more conducive to water absorption.

Second, hyphae increase the surface roughness of the assimilation branches of *H. ammodendron* and enhance its surface hydrophilicity. The epiphytic fungal hyphae on the assimilation branches of *H. ammodendron* can grow continuously and form thick, porous, felt-like hyphae, which is equivalent to increasing the surface roughness of the assimilation branches. According to the Wenzel model [40], $\cos\theta^{*}=r\cos\theta$ (r > 1), the surface roughness is r↑, rcos⁡θ↑, $\cos\theta^{*}$↑, $\theta^{*}$ decreases, and hydrophilicity increases. In another perspective, the thick hyphae on the surface of the assimilation branches greatly increase the surface area available for adsorbing atmospheric water vapor. This effectively enlarges the surface energy of the assimilation branches, allowing more atmospheric water vapor molecules to be adsorbed onto the surface of the hyphae. Compared to assimilation branches without hyphae, the number of water molecules adsorbed on the surface of the hyphae is much higher.

Third, hyphae extending through the cuticle layer facilitate water transport. The water molecules adsorbed on the surface of the hyphae permeate into the hyphae tube and flow inside as liquid water. The main component of fungal hyphae is chitin, which is a polysaccharide and polar molecule, and water molecules are also polar molecules, so the inner surface of the hyphae tube exhibits strong adhesion to water molecules, and the hyphae tube diameter is small (2.7–6.26 µm). According to the Kelvin Formula (1) mentioned below, the smaller the diameter of the tube, the greater the pressure of liquid flow in the tube. Therefore, liquid water flows under high pressure in the hyphae tube and can move quickly inside it. In this way, the water molecules adsorbed on the epiphytic hyphae will quickly enter the epidermal cells, without the hyphae penetrating the cuticle, avoiding the obstruction of water molecule transport by the hydrophobic cuticle. The hyphae of this fungus extend directly through the impermeable cuticle, transporting water much faster than water molecules can simply diffuse through the cuticle. Therefore, water molecules can be quickly transported to the endophytic hyphae in the epidermis of the assimilation branches of *H. ammodendron* by hyphae extending through the cuticle.

Fourth, the special position of endogenous hyphae is conducive to material exchange and transport. The endophytic hyphae of the assimilation branch of *H. ammodendron* were born in the gap between the epidermal layer and the hypodermis cells (Figure 3G,H), and were not inserted into the cytoplasm. On the one hand, the existence of a large number of endophytic hyphae could increase the area of material communication between the hyphae and the cell wall of the host, which was conducive to interaction. On the other hand, the endophytic hyphae do not insert into the protoplasm of the cell, the water is transported through the apoplast pathway, and the transmission speed is fast. The cell wall is relatively loose and is a structure with a large space for water and inorganic salt ions, so water molecules and inorganic ions can move freely in the cell wall, and water moves fast. The water does not enter the protoplasm in the cell, but is transmitted between the cell space or cell wall outside the protoplasm, which is an apoplast transmission, and the water transmission speed is fast. If the hyphae are located deep inside the protoplasm of the cell, the water transport must overcome the resistance of the protoplasts or solutes in the cytoplasm, the transport speed is slow, and the water will not be quickly transferred to other tissues or structures to carry out physiological activities in a timely manner, such as achieving rapid transportation to mesophyll cells and vascular sheath cells for photosynthesis. Thus, the endophytic hyphae growing between the epidermal layer and the hypodermis cells in the green part of the assimilation branches of *H. ammodendron* can accelerate water transport and meet the water demand of the mesophyll cells and vascular sheath cells of the assimilation branches of *H. ammodendron*, facilitating photosynthesis.

In short, the large number of endogenous hyphae in the assimilation branch of *H. ammodendron* increased the area of the hyphae that supply water to *H. ammodendron*, and the water supply speed was fast. In addition, the endophytic hyphae do not insert into the protoplasm of the cell; thus, the water supplied by endophytic hyphae directly enters the epidermal cell wall and is rapidly transported to the mesophyll cells and vascular sheath cells by the apoplast pathway for the synthesis of photosynthetic products.

Fifth, the hyphae can absorb atmospheric water vapor under low humidity. The porous thick external hyphae result in water absorption at low humidity. Because the hyphae are dense on the surface of the assimilation branches of *H. ammodendron*, and the gap between hyphae and hyphae is very small, it is equivalent to the formation of many capillary structures on the surface of the assimilation branches.

According to the Kelvin equation,
(1)$$RT\ln\frac{P_{r}}{P_{0}}=\frac{2\gamma M}{r\rho }$$

The structure of the capillary pores is equivalent to that of a concave surface. Applying the Kelvin equation to a concave surface, it can be seen that the pressure of the liquid and the saturated vapor in the capillary pores ($P_{r}$) are both smaller than that of the flat liquid ($P_{0}$). Therefore, under certain conditions, gas is more likely to condense into liquid inside than outside of pores. That is to say, under the same absolute humidity, it is easier to reach 100% relative humidity inside the capillary; in other words, the RH_0_ in the external environment is less than 100%, but the RHr in the capillary has reached 100%.

In the extreme arid areas of China, the RH_0_ of the atmosphere in the early morning is often 50–60%, and the RH_r_ in the pores of the hyphae on the surface of the assimilation branches of *H. ammodendron* may often reach 100%. Therefore, the porous thick hyphae formed on the surface of the assimilation branches of *H. ammodendron* can promote the absorption of atmospheric water vapor at a relatively low atmospheric relative humidity of 60%.

Sixth, the thick, felt-like hyphae reduce transpiration. The principle of stomata evaporation of lost water is as follows. When the stomata open, the atmospheric convection on the surface of the stomata will remove the water vapor in the leaf from the open stomata, so that the negative pressure of the water vapor is formed in the leaf, resulting in the liquid water in the catheter being vaporized and supplemented into the leaf. Since the convection at the stomata has always existed, the water vapor supplemented in the leaves will be continuously brought into the atmosphere from the stomata; that is, the water vapor continues to evaporate from the stomata. The stronger is the air convection in the open stomata, the greater is the transpiration. Based on this principle, as long as the air convection at the stomata is reduced, the stomata transpiration will be reduced. Thus, stomata depression, as well as the growth of hair on the epidermis or hyphae, will all reduce the atmospheric convection at the stomata and thus, will reduce the transpiration of H_2_O. The thick, felt-like hyphae structure on the assimilation branches of *H. ammodendron* can reduce the atmospheric convection at the open stomata, making it difficult to form negative water pressure in the assimilation branches. Naturally, the liquid water molecules in the assimilation branches can be slowed down to form gaseous molecules, and H_2_O molecules can be reduced to flow out from the stomata, especially when the external environment is dry, which greatly reduces transpiration and plays a role in water retention.

In summary, due to the absorption of atmospheric water vapor by a large number of hydrophilic hyphae on the epidermis of the assimilation branch of *H. ammodendron* and the cross-transmission of water through the hyphae without passing through the cuticle to avoid cuticle obstruction, coupled with the endogenous hyphae born between the epidermis and the hypodermis cells in the epidermis, the water molecules adsorbed by the superficial hyphae are quickly introduced into the epidermis and the hypodermis cells along the hyphae. Water is then quickly transported to the mesophyll cells and the vascular sheath cells via the apoplast transport pathway. In addition, the thick and porous felt-like hyphae structure can absorb atmospheric water vapor under low humidity, greatly reducing transpiration during drought. These characteristics explain that the hyphae on the surface of *H. ammodendron* assimilation branches can not only promote the absorption of atmospheric water vapor by *H. ammodendron* assimilation branches, especially under low humidity absorption, but can also reduce transpiration and play a role in water retention. As confirmed by our fluorescence experiment, the assimilation branches of *H. ammodendron* with hyphae on the surface could quickly adsorb atmospheric water vapor and be conducted rapidly at a relatively low atmospheric relative humidity of 60%, which was much faster than the rate for the assimilation branches without hyphae.

### 3.2. The Growth of Hyphae on the Assimilation Branches of Haloxylon ammodendron Is an Inevitable Ecological Process of Arid Ecological Adaptation

In early spring, the new shoots recently sprouted from the old branches of *H. ammodendron* do not require much water due to the limited number of new shoots. In addition, the soil moisture content is sufficient, and the growth needs of the *H. ammodendron* can be met solely by relying on the soil moisture content. There is no need to use atmospheric water vapor from the atmosphere, but the plant can rely solely on tender branches to absorb and supplement a small amount of atmospheric water vapor, which can fully meet the growth needs of plants. During extreme drought in the Ejina Banner area, we determined that the spring water potential of *H. ammodendron* was high (−3.39 MPa for mature new branches and −2.84 MPa for young new branches), and the water present on the branches was sufficient. The midsummer season, from July to mid-August, is extremely arid, with high temperatures, strong solar radiation, more evaporation, and less rainfall. At the same time, the new branches of *H. ammodendron* have grown and become more numerous. Many new branches have become mature branches, and the growth of mature branches consumes a large amount of water, while essentially using the same amount of soil water. In summer, the water potential of adult *H. ammodendron* branches in the Ejina Banner area is −5.71 MPa, while that of young branches is −3.48 MPa. Under this water condition, the tender branches absorb a large amount of atmospheric water vapor to supply the growth needs, and the mature branches can also absorb and utilize a small amount of atmospheric water vapor. However, due to the continuous absence of rain and the passage of seasons, it is not enough to rely on the water intake of young and mature branches. In order to adapt to the needs of this living environment, in late August, hypha began to grow on the surface of the assimilated branches of *H. ammodendron*, and through them, more atmospheric water vapor is absorbed and is better utilized. Our experimental Figure 4, Figure 5, Figure 6 and Figure 7 demonstrate this.

Secondly, in Lanzhou, during the same autumn, with good water conditions, there were no hyphae on the assimilate branches of *H. ammodendron*, and the assimilation branches of *H. ammodendron* in Lanzhou did not absorb atmospheric water vapor from the atmosphere. The hyphae grows on the assimilation branches of *H. ammodendron* only in extreme arid areas, where water conditions are scarce and the relative humidity of the atmosphere reaches 50–60%. We carried out fluorescent humidification experiments on the assimilation branches of *H. ammodendron* in Lanzhou in October, and the humidity was controlled at 85–90%. After 4 h of fluorescent humidification, only a few epidermal cells of the assimilation branches exhibited blue fluorescence, and additionally, only the epidermal cells near the stomata showed blue fluorescence on the slitting diagram [41]. In this case, the water does not reach mesophyll cells and vascular sheath cells for photosynthesis at all.

In contrast, when 60–65% of those without white hyphae in Ejina Banner area were fluorescent humidification, the entire epidermis of the assimilation branches of *H. ammodendron* could emit blue fluorescence after 0.5 h and the transverse images also showed that its epidermal cells emitted blue fluorescence, only less water entered the mesophyll cells; however, the assimilation branches with hyphae not only emitted blue fluorescence throughout the epidermis, but also water entered deeply into the assimilation branches, part of the water had entered mesophyll cells, and a small part had even reached vascular sheath cells (Figure 6 and Figure 7). The results show that the assimilating branches of *H. ammodendron* are easier to absorb water than those in Lanzhou, and those with hyphae have stronger water absorption capacity and faster transmission speed than those without hyphae.

The reason for this phenomenon may be related to the water status of the assimilation branches. Only under the condition of poor water condition, extreme drought, and late autumn season, *H. ammodendron* will grow hyphae on the assimilation branches, and generally, in semi-drought and drought regions, or under relatively good water conditions, it will not grow hyphae. The water condition of Lanzhou is good, so the deficiency value of *H. ammodendron* is low, so the assimilation branches do not need to absorb atmospheric water vapor from the atmosphere, and the hyphae do not grow on their assimilation branches, while in Ejina Banner, the water condition is poor and the water deficit value of *H. ammodendron* is high, so the assimilating branches can absorb atmospheric water vapor from the atmosphere to meet the water demand of the plants. In August, *H. ammodendron* plants were seriously short of water, and water absorption by assimilating branches alone was not enough to meet the water demands of the *H. ammodendron* plants. Thus, the assimilating branches cooperated with the fungal hyphae to promote water absorption by the hyphae to meet the water demand of the plants. Therefore, the *H. ammodendron* plant starts to sprout and abundance of hyphae on the assimilation branches.

In addition, there are atmospheric water vapor resources available in the extreme arid area. *H. ammodendron* is an ultra-xerophytic desert plant found in the extreme arid region in China. In the region, it is arid and hot in summer, with large temperature difference between day and night, and very little annual rainfall, with an average annual rainfall of 30–40 mm and only 7 mm, in some years. Although there is little rainfall in the growing season in the extreme arid areas, due to the large temperature difference between day and night during the growing season, high humidity levels of up to 50–60% can easily develop during the period from the early morning to the rising of the sun.

Appendix A shows a statistical table of the atmospheric relative humidity above 50% from May to September 2019 in Ejina Banner, an extremely arid region. According to Appendix A, there is an average of 3 h per day when the relative humidity of the atmosphere in Ejina Banner is greater than 50% during in the growing season. According to our experiment results, the atmospheric water vapor in this period can be utilized by the assimilation branches of *H. ammodendron* every day; thus, *H. ammodendron* can grow in this extreme drought condition.

Based on the above analysis, we believe that the growth of hyphae on the surface of the assimilation branches of *H. ammodendron* in extreme arid areas is an inevitable process for obtaining more water from the atmosphere, and it is a process evolving from refraining from utilizing atmospheric water vapor in favor of utilizing and making more and better use of atmospheric water vapor. The fact that hyphae can promote assimilating branches to absorb more water is the result of the ecological adaptation of *H. ammodendron* to the arid environment.

### 3.3. New Insights and Significance between White Powder Fungi and Haloxylon ammodendron

The white felt-like mycelium of the fungi growing on the assimilation branches of *H. ammodendron* comprises both epiphytic and endophytic hyphae. The conidia are thin and erect, growing out of the stomata. The conidia have two morphologies, with the apical tip of the primary conidium and the secondary conidium cylindrical. The cleistothecium is spherical or oblate, brown, and buried in the hyphae, and the appendages are filamentous. All these morphological characteristics are consistent with the genus *Leveillula*. Therefore, we believe that the fungus we studied, which grows on the assimilation branches of *H. ammodendron* with thick white felt-like hyphae, belongs to the genus *Leveillula*, family Erysiphaceae.

The thick, white, felt-like hyphae fungi growing on the assimilation branches of *H. ammodendron* were identified in the literature only once, as *L. saxaouli*, belonging to *Leveillula* genus, Erysiphaceae family [21], and it was considered that this fungus caused powdery mildew of *H. ammodendron* and was a pathogen. However, the white, felt-like hyphae fungus on the assimilation branches of *H. ammodendron* studied here was different from the *L. saxaouli* introduced in the literature in regards to the size of the cleistothecium and the size and morphology of the primary conidium. The size of the cleistothecium of the fungus identified by us was 15~26 μm in diameter, and the conidia were (17~33) × (5~8) μm, cylindrical, with a pointed tip, and did not enlarge into a ring near both ends (Figure 2H); the cleistothecium of the *L. saxaouli* introduced in the literature was 160~243 μm or 143~245 μm in diameter, and the conidia were (49~60) × (21~25) μm or (43~52) × (14~19) μm, with enlargement near both ends [21,22].

Comparing the two, the cleistothecium and conidium of the fungus we identified were smaller than those of the *L. saxaouli* introduced in the literature [21,22]; particularly, the cleistothecium was less than an order of magnitude, while the hyphae size of the two was on an order of magnitude, both being several microns, basically 3–7 μm. According to these results, we believe that the fungus we studied and the *L. saxaouli* identified in the literature are not be the same species, and it should belong to another species or another taxon under the same genus. As for whether or not this is the case, the fungus needs to be further identified by the gene sequence alignment of molecular biological methods.

Therefore, there may be more than one type of fungus that grows on the assimilating branches of *H. ammodendron* and forms thick, felt-like hyphae structures. Although all of them form thick, felt-like hyphae structures on *H. ammodendron*, the functions of different fungi may not be the same, just as the fungus we identified here, whose hyphae can promote the absorption of atmospheric water vapor through the assimilation branches of *H. ammodendron*, is not in a simple parasitic relationship with the host *H. ammodendron*. It was not as previously known that only one kind of fungus *L. saxaouli* grows on the assimilation branches of *H. ammodendron* with a thick, felt-like hyphae structure, and this fungus has been considered to be is a pathogen causing powdery mildew in *H. ammodendron*.

Of course, whether the thick, felt-like hyphae structure formed by *L. saxaouli* on the assimilation branches of *H. ammodendron* promotes the absorption of atmospheric water vapor or not requires further investigation and study.

In view of the above new knowledge, it is necessary to identify the fungi with white, felt-like hyphae on the assimilation branches of *H. ammodendron* in different regions at the molecular level in order to determine how many fungi form this white, felt-like hyphae structure on the assimilation branches of *H. ammodendron*. Whether or not there are multiple taxonomic groups and which areas of the fungi forming white, felt-like hyphae on the assimilated branches of *H. ammodendron* are beneficial to the absorption of atmospheric water vapor and which are pathogens all deserve further study in the future.

## 4. Experimental Site, Materials, and Methods

### 4.1. Experimental Site and Materials

The desertification steppe of Ejina Banner is taken as the research area. Ejina Banner in northwest China is an extremely arid region, with drought, low rainfall, significant evaporation, sufficient sunshine, large temperature differences, an abundance of wind and sand, and other climate characteristics. The average annual temperature is 8.3 °C, the average annual precipitation is 37 mm, the maximum annual extreme precipitation is 103.0 mm, and the minimum precipitation is 7.0 mm. The average annual evaporation is 3841.51 mm and the humidity is 0.01 mm.

*Haloxylon ammodendron*, a typical desert plant, was selected as the research object. In mid-September, several healthy 3–5 year old plants, with and without hyphae, were respectively selected as sample plants. The assimilation branches, with or without hyphae on the sample plants, were used as research materials for field fluorescence humidification experiments.

### 4.2. Experimental Methods

#### 4.2.1. Humidification Experiment

The humidification experiments were carried out on *H. ammodendron,* with or without hyphae, during the growing season in the study area. All field tests were conducted in 500 × 500 m plots. Two plants of *H. ammodendron,* at 3–5 years of age, with or without hyphae, were selected and covered with plexiglass for the humidification experiment. The connection between the glass plates of the humidification control room is sealed with transparent tape, and a 60 cm × 60 cm freely opening and closing glass door is left on the side for humidification and sampling. The soil surface of the control room is covered with plastic wrap to prevent water vapor or condensed water droplets from seeping into the soil during the humidifying. An ultrasonic humidifier is used to humidify the air in the control room. During the experiment, an ultrasonic humidifier was used to mist the water into ultrafine particles. A portable hygrometer (MicroLog PRO-EC750, Fourier Systems Ltd., Tel Aviv, Israel) was hung in the middle of the plant canopy in the control room to monitor the change of temperature and humidity in the control room in real time, locating the hygrograph away from the air outlet of the humidifier. A portable hygrometer was also hung at approximately the same position as the control plant. In order to avoid the high temperature caused by the sealed environment during the day affecting the normal physiological activities of the plant, humidification was carried out at night, without humidification or sealing during the day.

Two humidification methods are used, one being the use of ultra-pure water for humidification, which is achieved by adding ultra-pure water to the humidifier to measure the water potential and moisture content of the humidified assimilation branch, and the other being the use of FB fluorescent reagent for humidification to trace the water entry process. Humidifiers loaded with 0.1% (*w*/*v*) Fluorescent Brightener (FB) aqueous solution were used to humidify the sampled plants of *H. ammodendron*. FB is an apoplastic tracer [42] that binds to the polysaccharides of the cell wall and emits a strong, pale blue fluorescence under an excitation wavelength of 350 nm [43].

#### 4.2.2. Plant Water Potential

During the field experiment, a Psypro water potential meter (WESCOR, Inc., Logan, UT, USA) was used to measure the change of water potential in two kinds of *H. ammodendron* assimilation branches, with and without hyphae, under different humidification duration. Before measuring, clean paper was used to wipe off any moisture that may exist on the surface of assimilating branches or hypha. After collection, the assimilation branches were quickly cut and put into the Psypro dew-point water potential meter measuring chamber, sealed until it was balanced, and its water potential was measured using a probe. Three replicates were measured for each sample. The variation characteristics of the water potential of the assimilation branches, with and without hyphae, were compared during the ultrapure water humidification, and statistical analysis was conducted using the Wilcoxon test method in R software (R version 4.3.2).

#### 4.2.3. Moisture Content Determination

During the sampling, 6–10 the two types of new shoot assimilation branches of a similar length and size, one with hyphae and the other without hyphae, were cut off, any water on the surface of the assimilation branches and on the hyphae was wiped off with clean paper, and the water content of the two types of *H. ammodendron* assimilation branches was tested. The initial weight of new shoots (mass sample 0, g) was quickly weighed in the field with an electronic balance with an accuracy of 0.0001 g. After weighing, the assimilation branches were placed in an envelope containing a silicone desiccant, taken back to the laboratory, and baked at 105 ° C for 0.5 h, then at 85 ° C for 48 h, so that the branches and shoots were completely dry. After that, the dried and assimilation branches were weighed (mass sample n, g). According to the formula LWC_sample n_ (%) = (Mass_sample n_ − Mass_sample 0_) × 100%/Mass_sample 0_, the moisture content of the assimilation branches in different humidification periods was calculated. Each sample evaluated repeated three times for determination. The variation characteristics of the water content of the assimilation branches, with and without hyphae, were compared during the ultrapure water humidification, and statistical analysis was conducted using the Wilcoxon test method in R software (R version 4.3.2).

#### 4.2.4. Fluorescence Tracer Detection

The assimilation branches of the sample plants were collected from the experimental site, sealed in a self-sealing bag, stored in a refrigerator at 4 °C, and brought back to the laboratory. The epidermis was peeled off from some of the assimilation branches. The other branches were sectioned with a Small Plant Tissue Special Slicer MTH-1(DL Nature Gene Life Sciences, Inc., Tokyo, Japan) at a position in the middle of the branches. All of the preparations were then mounted in glycerol and observed under an Olympus BX53 fluorescence microscope (Olympus Corporation, Tokyo, Japan), under a bright field and under fluorescent conditions, with an excitation wavelength of 350 nm. Each sample was evaluated six times, including the control and the treated samples, using fluorescent humidifying at different times.

## 5. Conclusions

There may be another species of endo-filamentous white powder fungi growing on the assimilation branches of *H. ammodendron*. The thick, white, felt-like hyphae structure formed by this fungus on the assimilation branches can promote *H. ammodendron* to absorb atmospheric water vapor, which is a reliable ecological process to better adapt plants to arid environments, and an important survival strategy for other plants to grow or survive during drought in arid deserts. The relationship between the endo-filamentous white powder fungus and *H. ammodendron* does not occur merely through parasitism.

## Figures and Tables

**Figure 1 plants-13-01233-f001:**
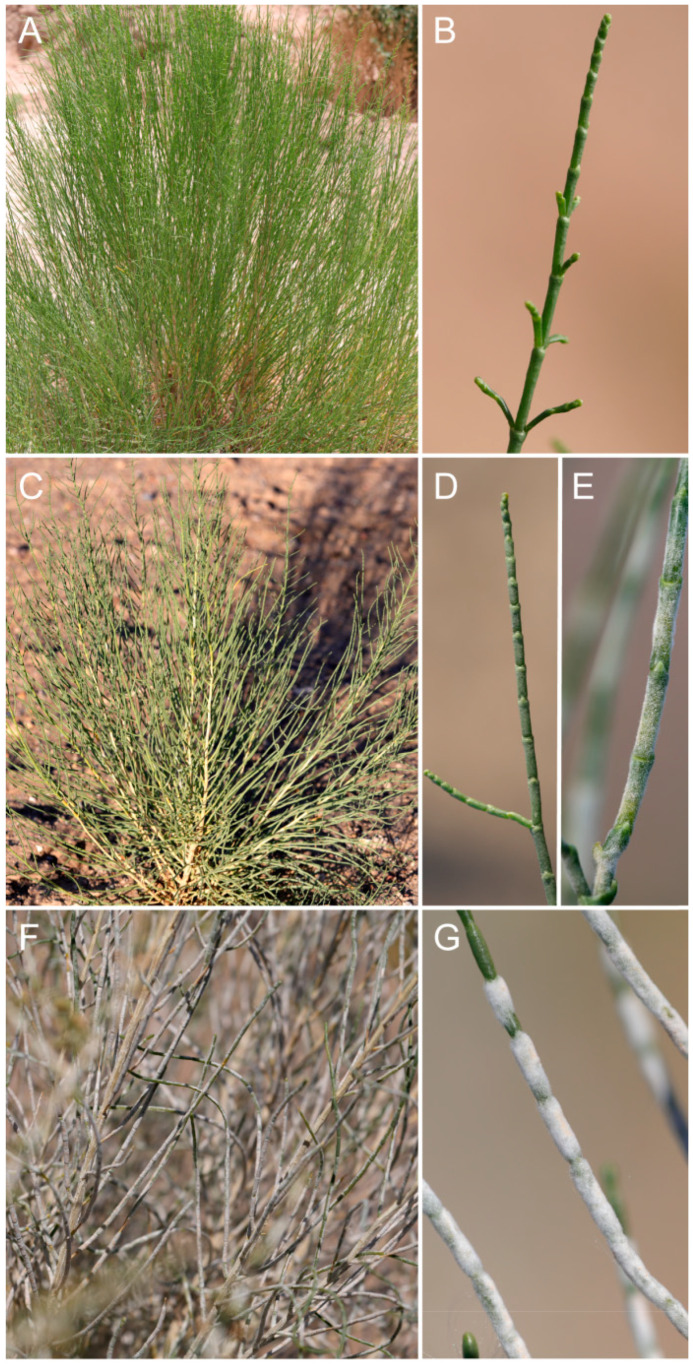
White hyphae on the surface of *Haloxylon ammodendron* assimilating branches ((**A**,**B**), no hyphae; (**C**–**E**), a small amount of white hyphae; (**F**,**G**) an abundance of white hyphae).

**Figure 2 plants-13-01233-f002:**
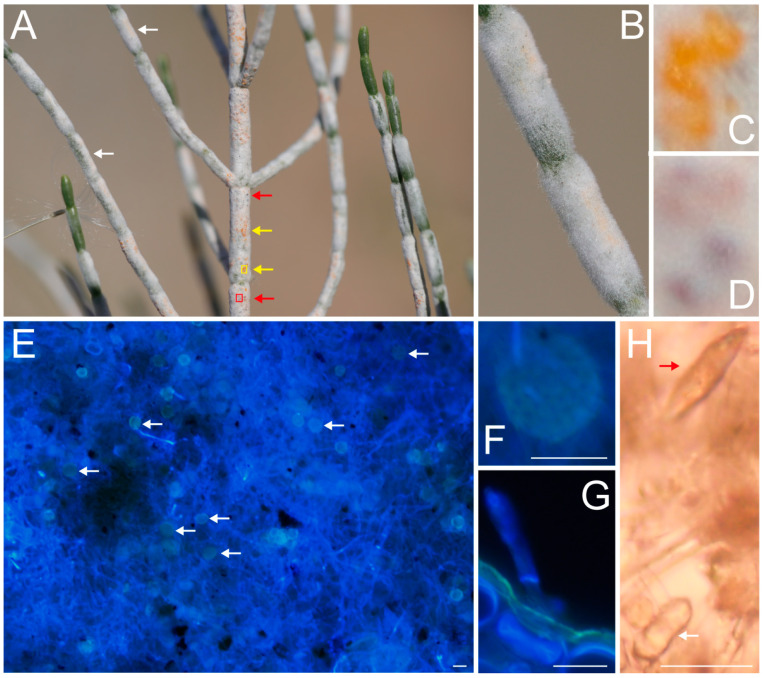
Infiltration laws of the fungus and its morphological identification. (**A**–**D**), the patterns of fungal infiltration of assimilation branches of *Haloxylon ammodendron*; (**A**), *Haloxylon ammodendron* assimilation branches been infiltrated by the fungus with white felt-like hyphae, the yellow arrows point light yellow edematous patches at the early stage of fungal infection, the red arrows point yellowish brown to brown cleistothecium at the late stage of fungal infection; (**B**), at the early stage of fungal infiltration, yellowish dots existing in the white hyphae on the assimilation branches; (**C**), enlarged light yellow edematous patches in the yellow box in (**A**); (**D**), enlarged yellowish brown to brown cleistothecium in the red box in (**A**); (**E**–**H**), the morphological identification of the fungus; (**E**), the cleistothecium embedded in the hyphae (indicated by the white arrow); (**F**), the enlarged cleistothecium; (**G**), the conidium growing out of stomata; (**H**), the red arrow indicates the primary conidium, and the white arrow indicates the secondary conidium. The line represents 20 µm.

**Figure 3 plants-13-01233-f003:**
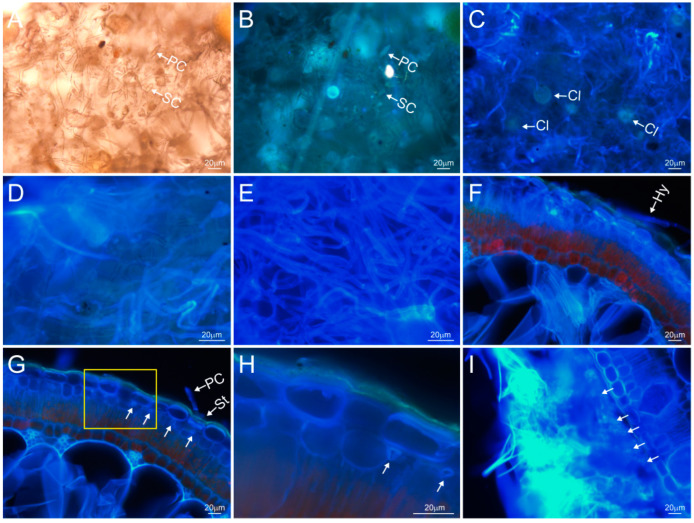
Structural characteristics of hyphae outside or inside the epidermis of assimilation branches of *Haloxylon ammodendron*. (**A**–**E**), hyphae surface images. (**A**), bright field optical image under natural conditions; (**B**), fluorescence image corresponding to (**A**); (**C**–**E**), fluorescence surface images after absorption of FB fluorescent reagent in the fluorescence humidification experiment; (**F**–**I**) cross-sectional fluorescence images after absorption of FB fluorescent reagent in the fluorescence humidification experiment; (**H**), the magnification at the yellow box of (**G**), and the white arrow indicates the cross section of the hyphae. PC, primary conidium; SC, secondary conidium; Cl, cleistothecium; Hy, hyphae; St, stomata.

**Figure 4 plants-13-01233-f004:**
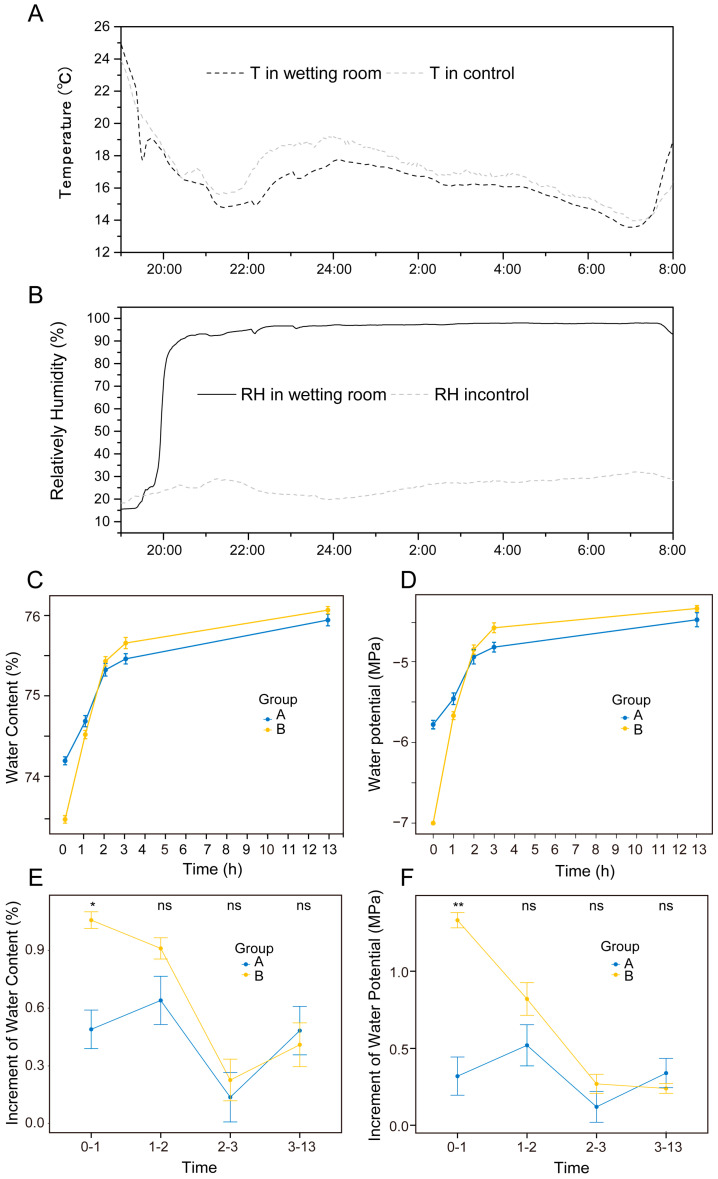
Comparison of water content and water potential of assimilation branches, with and without hyphae, during the ultrapure water humidification. (**A**,**B**), the temperature and relative humidity in the humidification experiment room and in the natural atmospheric environment; (**C**), change in water content (%); (**D**), change in water potential (MPa); (**E**), change in water content during a time period (%); (**F**), change in water potential during a time period (MPa). Data were organized by Office Excel 2016, and statistical analysis was performed using R language (R version 4.3.2) software employing the Wilcoxon test method: *, *p* < 0.05; **, *p* < 0.01; and ns, no statistically significant difference. In (**C**–**F**), A in the legend indicates assimilation branches without hyphae, and B indicates assimilation branches with hyphae.

**Figure 5 plants-13-01233-f005:**
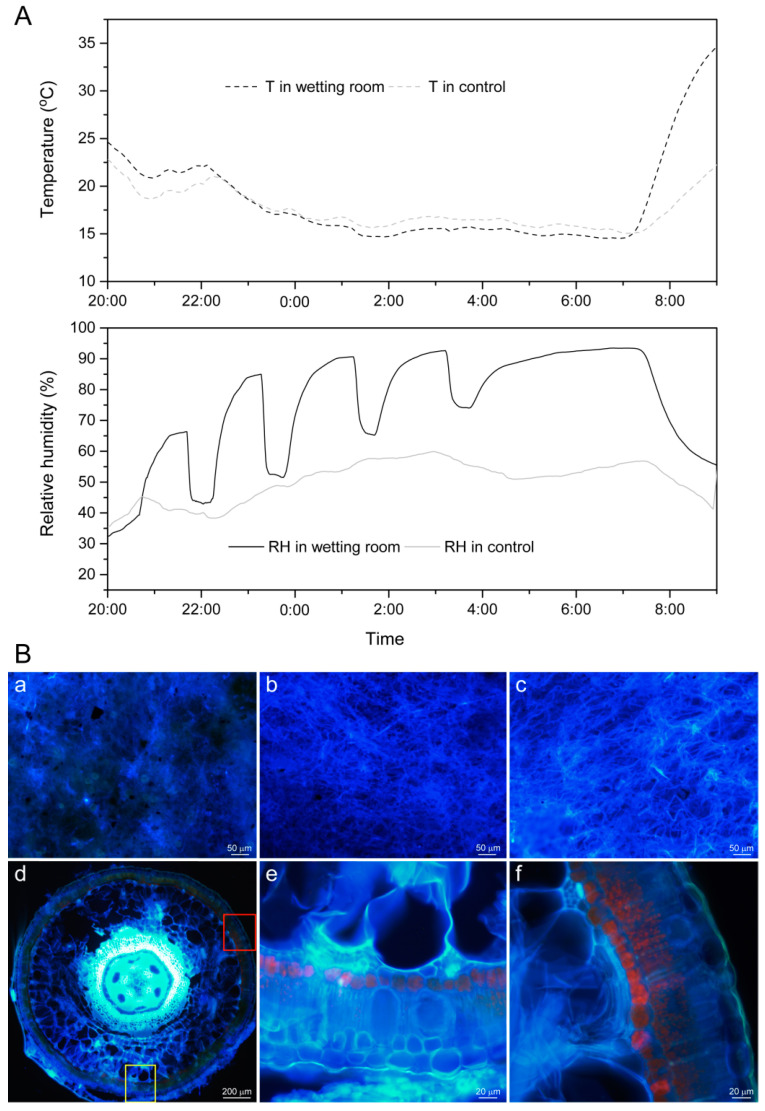
Comparison of water absorption of fluorescence humidification between assimilation branches, with hyphae and without hyphae. (**A**) Comparison of indoor and outdoor temperature and humidity for fluorescent humidification. (**B**) Comparison of the fluorescence emitted by hyphae on *Haloxylon ammodendron* assimilation branches under different fluorescence humidification durations. The duration of fluorescence humidification for (**a**), (**b**), and (**c**) was about 0.5 h, 1.5 h, and 3 h, respectively. (**d**) A complete cross-sectional fluorescence comparison of hyphae or hyphae-free parts of the same assimilation branch; (**e**) enlargement of the yellow box in the area with hyphae in (**d**); (**f**) magnification of the red box at hyphae-free site of (**d**).

**Figure 6 plants-13-01233-f006:**
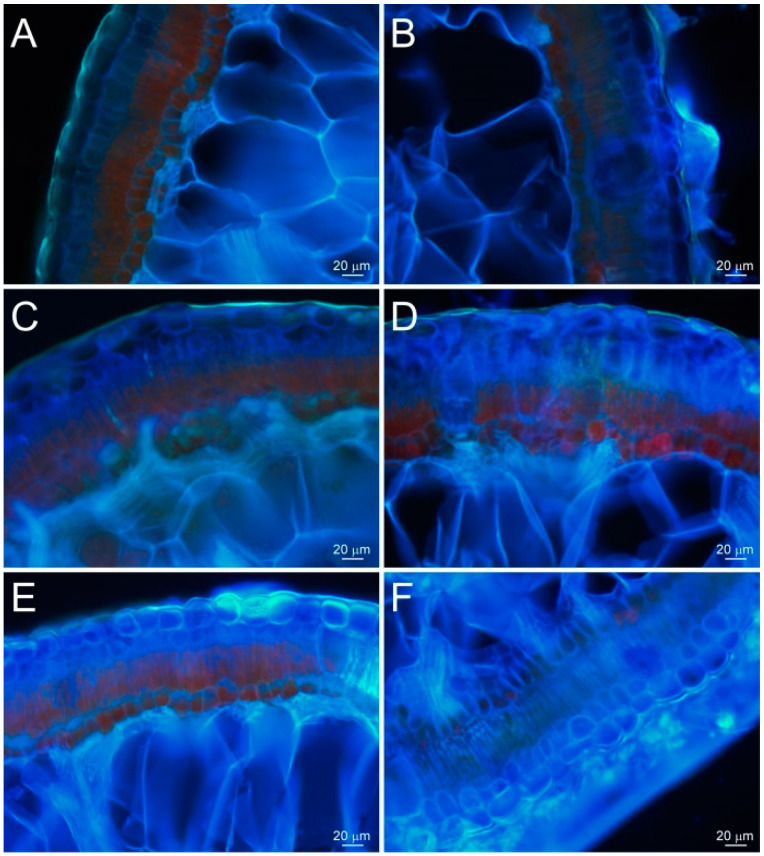
Comparison of absorption of atmospheric water vapor by assimilation branches of *Haloxylon ammodendron,* with hyphae and without hyphae, under different fluorescence humidification durations. (**A**,**C**,**E**) were used for assimilating branches without hyphae for about 0.5 h, 1.5 h, and 4.5 h, respectively. (**B**,**D**,**F**) were used for assimilating branches with hyphae at about 0.5 h, 1.5 h, and 4.5 h, respectively.

**Figure 7 plants-13-01233-f007:**
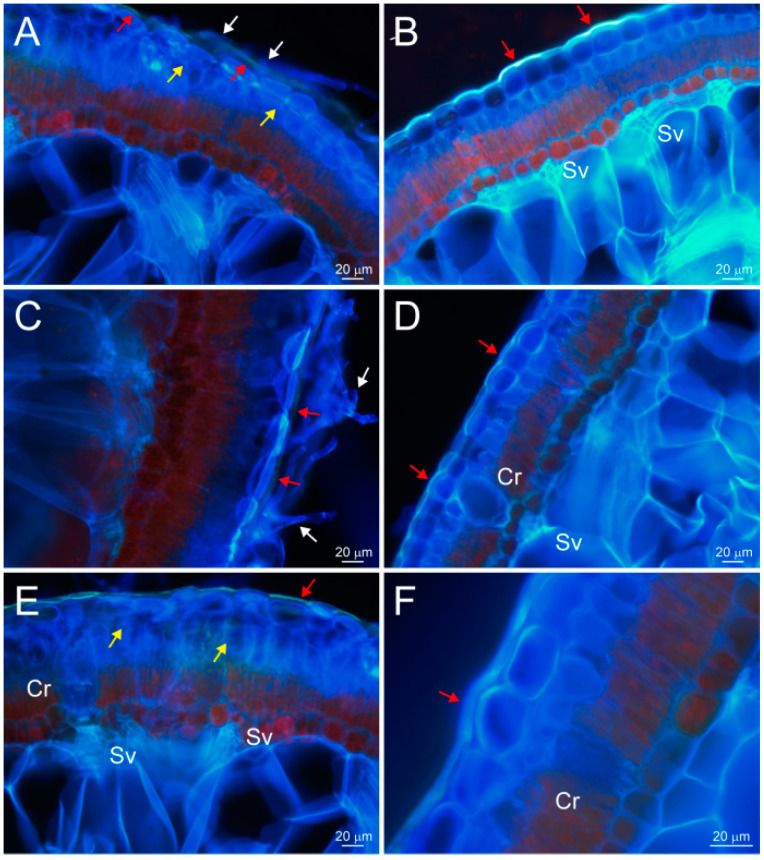
Comparison of water transport pathways between the epidermis of assimilation branches of *Haloxylon ammodendron,* with hyphae and without hyphae, under the same humidification conditions. (**A**,**C**,**E**) Transport routes of water in the epidermis with hyphae in assimilation branch after humidification for 1.5 h. (**B**,**D**,**F**) Transport routes of water in the epidermis of no-hyphae branches for about 3–4.5 h humidification. The white arrow indicates the hyphae, the red arrow indicates the cuticle, and the yellow arrow indicates the cross section of the hyphae in the epidermis. Sv, small vascular bundle; Cr, crystalline cells.

## Data Availability

Data are contained within the article and Appendix A.

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
