# Peer review of "Fungal Hyphae on the Assimilation Branches Are Beneficial for Haloxylon ammodendron to Absorb Atmospheric Water Vapor: Adapting to an Extreme Drought Environment"

_plants, 2024, doi:10.3390/plants13091233_

Round 1
Reviewer 1 Report
Comments and Suggestions for Authors
In the manuscript “ Fungal Hyphae on the Assimilation Branches are Beneficial for Haloxylon ammodendron to Absorb Atmospheric Water Vapour: Adapting to Extreme Drought Environment, the authors analyse endophytic and epiphytic fungi in desert plants, which is an interesting subject of research due to the scarce papers published. It will be interesting for a broad auditorium.
The paper represents a comprehensive investigation of the symbiosis between some fungi and the active photosynthetically epidermal zones of young branches in a plant growing in extremely desert ecosystems. The content represents a lot of work and a lot of good analysis, and it is a substantial advance on the state of knowledge of this type of symbiosis
It is very interesting the amount of results that has allowed them to demonstrate that in the growth of the fungus are involved three types of hyphae that can achieve the effect of rapid water absorption and transfer to inside the branches, but also realize the absorption of atmospheric water vapour under low humidity in the atmosphere. The fungi are not developed when environmental conditions are mild for the plants (i.e. rainy spring), consequently this is a very interesting type of symbiosis between fungi and aerial parts of vascular plants and it is the demonstration of a rare example of drought ecological adaptation.
The study is in general well written. The data analysis, the methods and results were adequate and precisely described and graphically documented.
Although:
1. The abstract would need minor revision to improve better understanding for the non-expert readership.
2. The discussion is unnecessarily long and sometimes repetitive in its arguments, it needs to be synthesized.
3. There are small spelling errors that you should correct when revising the manuscript. An important one is in the title: change Vapor for Vapour
I have a main issue that deserve further thought:
My main concern is that, although the research carried out is so interesting and novel, they have not identified molecularly the species, or at least the genus, of the fungus studied. They refer to an ascomycete that others have found in this article: Zhao, Z.Y. Xinjiang powdery mildew flora. 1979. Xinjiang People's Publishing House.
It would also be necessary to know if the different types of hyphae found with different functions are from a single fungus or if there are several taxa.

Comments on the Quality of English Language1. The abstract would need minor revision to improve better understanding for the non-expert readership.
2. The discussion is unnecessarily long and sometimes repetitive in its arguments, it needs to be synthesized.
3. There are small spelling errors that you should correct when revising the manuscript. An important one is in the title: change Vapor for Vapour
Reviewer 2 Report
Comments and Suggestions for Authors
In my opinion, this study is incomplete. There is no solid statistical evidence to support the symbiosis between Haloxylon ammodendron and Leveillula saxaouli. The effects of the fungus on the plant's ecophysiological parameters, growth and development are ignored.
Round 2
Reviewer 1 Report
Comments and Suggestions for Authors
The authors have answered to most of my comments and have shortened the Discussion and clarified the conclusions.
However,
I think that the identification of this fungus is Leveillula saxaouli has not been demonstrated! by the authors. At any time, they did not validate the identification of the fungus at the specific range, only by what was said in the previous literature of more than 20 years ago.
It is difficult to accept your answer to my COMMENT 2: “We did not identified the species studied at the molecularly level is because the focus of this paper is to evaluate leaf surface interactions and potential foliar water uptake processes, while neither to characterize the symbiosis between the fungus and the plant H. ammodendron at a microbiological level nor at a plant physiology level”
I am sorry, but you have not understood that this is not your purpose in the manuscript: this fungus is a key organism in all the processes you have been able to study. These are interesting and novel results, but they lack the most important basis: you have not identified adequately the fungus that you are studying, and this is not acceptable from a taxonomic, botanical scientific point sof view. You only refer its identification to papers written by other authors over 20 years ago. You, I and other scientists cannot confirm that Leveillula saxaouli is the fungus you have studied.
The contradiction is that you have correctly identified the vascular plant on which the fungus grows!
OTHER OBSERVATIONS:
Lines16-18
“This thick felt-like hyphae layer facilitates to adsorb atmospheric water vapour onto the surface of hyphae or assimilating branches, enabling H. ammodendron can capture atmospheric moisture even at low humidity.”
Suggestion of writing: This thick felt-like layer of mycelial hyphae facilitates the adsorption of atmospheric water vapor on the surface of the hyphae or assimilating branches, allowing H. ammodendron to capture atmospheric moisture even at low humidity.
Comments: When the name of a species is written in the legends of figures or tables, the generic name should be included in full. And, when the scientific names are in a separate section of the text, they should be written in normal format, not in italics.
Lines
60 Change Chenoodiaceae to Chenopodiaceae
136 H. à Haloxylon
139 H. -à Haloxylon in normal type
165 H. -à Haloxylon
205 H. à Haloxylon
237 H. -à Haloxylon
272 H. -à Haloxylon en redondas
309 include: branches of Haloxylon ammodendron
482 H. -à Haloxylon en redondas
3.2
573 H. -à Haloxylon en redondas
786 H. -àHaloxylon
Conclusions:
Suggestion of writing: Fungal hyphae growing on the assimilating branches of Haloxylon ammodendron can promote them to absorb atmospheric water vapor, which is a reliable ecological process to better adapt to arid environments, and an important survival strategy for other plants to grow or survive from drought in arid deserts. The relationship between the white powder fungus and H. ammodendron is not merely through parasitism.
578-581. So previously, this phyllosphere fungus, which grewows on as-578 similating branches of H. ammodendron with epiphytic and endophytic hyphae, was considered as a pathogen, believed to cause white powder disease. It was classified as a species of white powder fungus and named Leveillula saxaouli.
IT IS NOT TAXONOMICALLY AND SCIENTIFICALLY ACCEPTED, AND EVEN LESS SO IN A DISCUSSION
Reviewer 2 Report
Comments and Suggestions for Authors
I believe, further, that there are not enough experimental arguments to prove the authors' conclusion.
The results for "Moisture Content Determination" are not provided.
No statistically interpreted data.
Round 3
Reviewer 1 Report
Comments and Suggestions for Authors
The authors have answered to most of my comments and have shortened the Discussion and clarified the conclusions.
They also change adequately the comments about the taxonomy of fungal species.
I agree with the inclusion of two new authors Zhao Xin and Wang Haixin that have improved the content of the manuscript.
Reviewer 2 Report
Comments and Suggestions for Authors
The content of the article has been improved.